# Recent Approaches towards Control of Fungal Diseases in Plants: An Updated Review

**DOI:** 10.3390/jof7110900

**Published:** 2021-10-25

**Authors:** Nawal Abd El-Baky, Amro Abd Al Fattah Amara

**Affiliations:** Protein Research Department, Genetic Engineering and Biotechnology Research Institute (GEBRI), City of Scientific Research and Technological Applications (SRTA-City), New Borg El-Arab City, Alexandria 21934, Egypt

**Keywords:** plant fungal diseases, phytopathogenic fungi, biocontrol, botanical fungicides, microbial fungicides, agronanotechnology, nanoparticles, ghost techniques

## Abstract

Recent research demonstrates that the number of virulent phytopathogenic fungi continually grows, which leads to significant economic losses worldwide. Various procedures are currently available for the rapid detection and control of phytopathogenic fungi. Since 1940, chemical and synthetic fungicides were typically used to control phytopathogenic fungi. However, the substantial increase in development of fungal resistance to these fungicides in addition to negative effects caused by synthetic fungicides on the health of animals, human beings, and the environment results in the exploration of various new approaches and green strategies of fungal control by scientists from all over the world. In this review, the development of new approaches for controlling fungal diseases in plants is discussed. We argue that an effort should be made to bring these recent technologies to the farmer level.

## 1. Introduction

The vast majority of known fungal species are strict saprophytes; only very few species (less than 10% of identified fungi) can colonize plants. Phytopathogenic fungi represent an even smaller fraction of these plant colonizers. Yet, phytopathogenic fungi are the key causative agent among phytopathogens for devastating crop plant epidemics, besides causing persistent and substantial losses in crop yield annually. Thus, phytopathogenic fungi are battled by scientists, plant breeders, and farmers equally due to these economic factors [1,2].

Commercial agriculture depends mainly on the application of chemical fungicides to protect crop plants against fungal pathogens by destroying and inhibiting their cells and spores. However, their easy application and low cost result in their overuse or repeated applications [3]. This overuse or misuse of fungicides has led to toxic effects on beneficial living systems, human and animal health, and the environment. Moreover, the emergence of resistant strains of fungal phytopathogens makes plant fungal diseases become increasingly challenging to treat. Accordingly, development of healthy, non-toxic, and eco-friendly alternate approaches (green strategies of fungal control) to chemical and synthetic fungicides is very helpful in the control of plant fungal infections [3,4,5,6]. These safe and effective alternative control means against plant fungal diseases include biological control of phytopathogenic fungi [7], microbial fungicides [7,8], botanical fungicides [9], agronanotechnology [10,11], and fungal cell deactivation and evacuation using ghost techniques [12].

In this article, we reviewed biocontrol, biofungicides, microbial fungicides, botanical fungicides, agronanotechnology, and fungal cell deactivation and evacuation using ghost techniques that represent recent, safe, and effective alternative control means (green strategies of fungal control) against plant fungal diseases that have been reported in the scientific literature but have not yet been properly introduced to farmers.

## 2. Biological Control of Phytopathogenic Fungi

### 2.1. Biofumigation (Biological Soil Disinfection)

This control method is based on fresh organic material incorporation in the soil and then its plastic tarping [13]. It employs organic material fermentation in soil under plastic cover to produce anaerobic conditions and toxic metabolites leading to inactivation of phytopathogenic fungi. The technique was further developed and classified by Lamers et al. (2004) [14] to biofumigation using distinctive plant species comprising well-known toxic molecules, and biodisinfection by organic substances that produce anaerobic conditions for inactivation of phytopathogenic fungi. 

Plant species of the *Alliaceae* (onion) family produce toxic molecules that directly or indirectly affect fungal plant pathogens. For example, garlic and onion tissues degradation leads to sulfur volatiles’ (zwiebelanes and thiosulfinates) release, which are then converted into biocidal disulfides against phytopathogenic fungi [15].

Blok et al. (2000) [13] reported the control of soil-borne phytopathogenic fungi (*Fusarium oxysporum* and *Rhizoctonia solani*) by integrating fresh organic matter such as cabbage or ryegrass in soil followed by plastic tarping. These methods represent promising substitutes for banned methyl bromide disinfection, which harms the human respiratory and central nervous. 

A recent laboratory and greenhouse study was carried out in Egypt which involved use of biofumigation with Indian mustard (*Brassica juncea*) to control *Rhizoctonia solani* infection of the common bean. They used *Brassica juncea* as fresh and dry plants, methanol extract, or seed powder and meal [16].

### 2.2. The Use of Antagonistic Microorganisms in Suppressive Soils

Phytopathogenic fungi can be controlled by adding suppressive soil that comprises antagonistic microorganisms (microbes antagonistic to phytopathogenic fungi) to natural pathogen-conducive soil. This added suppressive soil results in fungal pathogen and plant fungal disease suppression [17]. Various antagonistic microorganisms were identified in suppressive soils, but fungi were the dominant microbes among them that have the ability to suppress pathogens and diseases. For instance, to control papaya root rot caused by *Phytophthora palmivora,* papaya seedlings were planted in suppressive virgin soil and added to holes in the infected plantation soil [7]. This control protocol was followed for protecting papaya roots during early stages of growth (seedlings), because papaya roots are susceptible to *Phytophthora palmivora* only when the plant is young. Virgin soil used was collected soil from land that had never been used for growing papaya, and thus was generally free from *Phytophthora palmivora* infection, which occurs only in replanting fields. About 42% of papaya seedlings planted in holes in the infected plantation soil without virgin soil died three months after planting, while all of those planted in holes in the infected plantation soil with virgin soil survived. Another example is control of the phytopathogenic fungus *Fusarium oxysporum* that causes wilt disease using suppressive soils [18]. Other studies involved analysis of *Fusarium* wilt suppressive soils from Chateaurenard and identified new bacterial and fungal genera in these soils that play a key role in suppression of *Fusarium* wilt [19].

In some cases, only monocultures of the same crop in a pathogen-conducive soil will reduce plant fungal disease after years of severe infection, as antagonistic microflora to the fungal pathogen will increase with passing time [20]. An example of this is monoculture of cucumber or wheat that reduces infections of cucumber damping-off and wheat take-all, respectively, caused by *Rhizoctonia* [7]. Other studies argue that intercropping or the simultaneous cultivation of various species of crops exceeds monocropping in disease control [21,22]. 

A third example of the use of soil suppressiveness in controlling phytopathogenic fungi is cultivation of proper crops as soil amendments [23,24]. These crops (mostly cruciferous vegetables) offer resident antagonistic microflora in the soil for biocontrol of pathogens. For instance, biocontrol of *Sclerotinia sclerotiorum* that causes lettuce drop was achieved by broccoli incorporation in the soil [25].

### 2.3. Microbial Control of Phytopathogenic Fungi

The governmental reviews of the safety of chemical synthetic fungicides result in special interest in microbial fungicide development and use. There are two comprehensive types of microbial fungicides used for biocontrol of plant fungal diseases. The first type directly interacts with the target fungal plant pathogens via various mechanisms such as parasitism, antibiotics production against target pathogens, or even competition with soil-borne phytopathogenic fungi for food, water or space because they are occupying the same ecological niche. The second type indirectly affects target fungal plant pathogens by inducing plant resistance against virulent pathogens. This inducer type can be a low virulent plant-pathogen strain, another microbial species, or their natural products. 

In general, microbial antagonists used for biocontrol of plant fungal diseases have multiple mechanisms involved in their action. *Trichoderma* species, for example, act against soil-borne phytopathogenic fungi through parasitism, production of antibiotics and enzymes that degrade the fungal cell wall, competition for nitrogen or carbon, and also by producing auxin-like compounds causing plant growth promotion [26,27]. 

Various microbial antagonists have been developed and commercialized to be used against soil-borne phytopathogenic fungi that cause diseases in the above-ground plant parts. Species of *Trichoderma*, for example *Trichoderma harzianum*, are one of the most widely used microbial antagonists for biocontrol of plant fungal diseases caused by *Fusarium, Rhizoctonia*, and many soil-borne phytopathogenic fungi [28]. Another microbial fungicide is *Coniothyrium minitans*, which is used for biocontrol of infections of lettuce, oilseed rape, brassicas, beans, and carrots caused by *Sclerotinia sclerotiorum* [29]. The bacterium *Paenibacillus jamilae* HS-26 was reported to have potent antagonistic effects (inhibiting mycelial growth of fungi) on multiple soil-borne fungal pathogens via releasing extracellular antifungal metabolites and the synthesis of hydrolytic enzymes [30]. Formulations of *Streptomyces cellulosae* Actino 48 that produces chitinase were reported to control *Sclerotium rolfsii* causing peanut soil-borne diseases [31]. 

Formulations of microbial fungicides include liquid suspensions, granules, or dusts, which are applied in the soil just before cultivation or directly to plant roots. They can also be formulated as conventional sprays and applied on harvested fruits, plant stems, or leaves. Moreover, unique application methods have been developed such as honey bees’ delivery during pollination [32]. Bees usually carry *Monilinia vaccinii-corymbosi* (a phytopathogenic fungus that causes mummy blueberry disease) between the flowers of blueberry during pollination. At the same time, the bees can act as ‘flying doctors’ and deliver the bacterial fungicide *Bacillus subtilis* to the flowers of blueberry to suppress the disease [33]. Additionally, the endophytic bacterium *Bacillus mojavensis* was reported to be fungicidal against various phytopathogenic fungi including *F. oxysporum*, *R. solani*, and *Sclerotinia sclerotiorum* [34]. 

### 2.4. Botanical Fungicides

Many published studies reported that plant extracts exhibit significant antifungal activities *in vitro*. Unfortunately, agar diffusion assays were unsuitably used to detect these activities of plant extracts, while several antifungal compounds found in plant extracts are relatively non-polar and consequently do not diffuse well in agar [35]. The obtained results were also highly variable from one laboratory to another due to the variation in factors that affect agar diffusion. 

Numerous essential oils of plants have the ability to suppress fungal infections that initiate and develop during and after crop harvest and thus extend the shelf-life of stored vegetables and fruits [36,37]. They can also inhibit production of mycotoxins by some species of fungi that cause postharvest decay of stored fruits [38].

Some plants are able to produce various antimicrobial agents (natural or botanical fungicides) to protect themselves against several plant fungal diseases [39]. These natural fungicides, such as phytoanticipins and phytoalexins differ in their structure, molecular weight, functions, and classification [40]. Secondary metabolites produced by plants which can act as natural fungicides for control of phytopathogenic fungi include phenolics, fatty acids, flavonoids, alkaloids, glycosides, terpenoids, and tannins. 

Plant extracts usually have the advantage of comprising various chemicals (antifungal compounds) mixed together, and possibly will work in a synergistic manner against phytopathogenic fungi [41]. Additionally, diverse mechanisms of antifungal activity employed by these mixtures of compounds could result in a decrease in the resistance of fungal phytopathogens. The following are some examples of commercially available botanical fungicides besides several metabolites produced by plants that displayed effective antifungal activities against fungal phytopathogens in vivo.

#### 2.4.1. Milsana

Milsana is a botanical fungicide extracted from the *Reynoutria sachalinensis* plant with ethanol to be used for the reduction of some infections that affect greenhouse-grown plants, principally powdery mildew [42]. It is mostly applied as a preventative agent rather than a treatment. The control mechanisms that Milsana employs to suppress powdery mildew disease of wheat include its antifungal activity as well as inducing resistance of the plant. To effectively reduce powdery mildew that affects young seedlings in glasshouses by about 97%, this botanical fungicide should be applied as spray to run-off once at 48 h before planting [43,44]. Milsana stimulates resistance and the natural immune system of the plant via acting as a natural elicitor of phytoalexins, which are antimicrobial compounds synthesized and accumulated by plants in hypersensitive tissues as a response to pathogen infection [45].

#### 2.4.2. Jojoba Oil

This vegetable oil has been extracted from the bean of jojoba. It can effectively control powdery mildew fungal disease in grapes and ornamental plants [45,46]. Moreover, its stability, even at elevated temperatures, presents jojoba oil as a broadly functional fungicide in approximately all climatic conditions [45,46]. This botanical fungicide is sprayed at a final concentration of about 1% [45]. 

#### 2.4.3. Plant Essential Oils

Plant essential oils are concentrated liquids comprised of volatile chemical compounds from plants. They can be referred to as volatile oils or simply as the oil of the plant from which they were extracted. They are generally extracted by distillation, expression, solvent extraction, or resin tapping. They have many applications, such as flavoring foods and drinks, their use in cosmetics, perfumes, soaps, etc. and aromatherapy (alternative medicine) in which healing effects are attributed to aromatic compounds [47].

Edris and Farrag [48] evidenced that essential oils from sweet basil and peppermint in addition to their key aroma constituents (menthol in case of essential oil of peppermint and linalool/eugenol in case of basil oil) have antifungal abilities against some phtopathogenic fungi, including *Rhizopus stolonifer*, and *Sclerotinia sclerotiorum*.

Additionally, Calocedrus macrolepis essential oil and its components was reported to have an antifungal effect on fFusarium oxysporum, Fusarium solani, Rhizoctonia solani, Pestalotiopsis funerea, and Colletotrichum gloeosporioides [49].

In another study, essential oils extracted from twenty five different medicinal plant species were confirmed to have inhibitory effects (inhibiting mycelial growth of fungi) on six vital toxinogenic and pathogenic species of fungi (*Fusarium verticillioides,*
*Fusarium oxysporum*, *Aspergillus fumigatus*, *Aspergillus flavus,*
*Penicillium brevicompactum,* and *Penicillium expansum*) [50]. 

Elgorban et al. (2015) [51] demonstrated that essential oils extracted from *Eucalyptus globulus*
*Labill*, *Nigella sativa* L., and *Allium cepa* L. have antifungal activity against *Rhizoctonia solani, Sclerotinia sclerotiorum*, *Fusarium oxysporum*, *Fusarium verticillioides*, and *Fusarium solani*).

Elshafie et al. (2016) [52] characterized the chemical composition of three essential oils extracted from *Majorana hortensis*, *Verbena officinalis*, and *Salvia officinalis*, and their antifungal activity was confirmed against *Colletotrichum acutatum* and *Botrytis cinerea.* The chemical structure of studied essential oils was mostly composed of monoterpene compounds and all oils belonging to the chemotype carvacrol/thymol. A more recent work has been conducted by Elshafie et al. (2019) [53] to study the fungicide effect of essential oil from *Solidago canadensis* L. and its effect on some postharvest phytopathogenic fungi such as *Aspergillus niger*, *Botrytis cinerea*, *Monilinia fructicola*, and *Penicillium expansum* was confirmed. Two essential oils derived from *Origanum heracleoticum* L. and *O. majorana* L. were reported to have in vitro antifungal activity against some postharvest phytopathogens (*Aspergillus niger, Penicillium expansum*, *Botrytis cinerea*, and *Monilinia fructicola*) [54].

In a recent study, Perczak et al. (2019) [55] used certain essential oils as antifungal compounds against *Fusarium culmorum* and *Fusarium graminearum* to constrain their growth and mycotoxins production in wheat grain. 

However, regardless of the effective antifungal activities of essential oils against various phytopathogenic fungi reported many times in the scientific literature, their application in agriculture continues to be unexpectedly scarce [56].

#### 2.4.4. Cinnamaldehyde

Since first isolation of cinnamaldehyde from the essential oil of cinnamon in 1834, it has been chemically synthesized to be used for crop protection. It can effectively control the phytopathogenic fungi *Sclerotinia homoeocarpa*, *Verticillium fungicola*, and *Fusarium moniliforme* that cause dollar spot of turfgrasses, dry bubble on *Agaricus bisporus*, and pitch canker infection, respectively [45]. It was reported that cinnamon has antifungal activity against fungi, causing wood decay [57].

Cinnamaldehyde was also shown to exhibit antifungal effects on wood-rot fungi [58]. Furthermore, Khan et al. (2011) [59] demonstrated that cinnamaldehyde has the highest antifungal effects among other key components (citral, eugenol, or geraniol) of essential oils. It showed the highest inhibitory effect on fungal growth of *Trichophyton rubrum* and *Aspergillus fumigates* besides negatively affecting their virulence factors and ultrastructure of hyphae. 

Cinnamaldehyde formulated as wettable powder is commercially available and sold under the names VERTIGO and Proguard fungicides. This formulation is water insoluble and promptly degraded in soil without causing environmental hazard or harm to non-target organisms.

#### 2.4.5. Phenolic Compounds

Many studies reported that medicinal plants produce phenolic compounds with antifungal activity [9,60,61,62,63]. These plant phenols are classified as simple phenolics, anthraquinones, flavonoids, and coumarins. It was demonstrated that curcuminoids (polyphenolic pigments) purified from *Curcuma longa* rhizomes exhibit antifungal action against several phytopathogenic fungi in in vivo studies [9,61,62]. The rhizomes of *C. longa* extracted with methanol comprise three curcuminoids, principally curcumin, bisdemethoxycurcumin, and demethoxycurcumin. Demethoxycurcumin displayed the highest antifungal effects among *C. longa* curcuminoids on late blight of tomato and blast of rice, followed by curcumin. Furthermore, *C. longa* curcuminoids could efficiently inhibit spore germination as well as growth of mycelia in *Colletotrichum coccodes*, a phytopathogenic fungus that causes anthracnose on red pepper. Nevertheless, in vivo studies revealed that *C. longa* curcuminoids showed no or little antifungal effects on phytopathogenic fungi causing powdery mildew on barley, blight of rice sheath, leaf rust of wheat, and gray mold of tomato. Choi et al. (2008) [60] also reported that the fruit rinds of *Myristica malabarica* extracted by methanol comprise malabaricones (diarylnonanoids), which exhibit antifungal activity against phytopathogenic fungi.

#### 2.4.6. Alkaloids Compounds

These compounds have nitrogen in their rings, and are classified based on their heterocyclic ring to isoquinoline alkaloids, pyridine alkaloids, and indole alkaloids. They can play a role in plant defense against pathogens and herbivores. Concerning their medicinal properties, some alkaloids are analgesic while others are cardiac or respiratory stimulants. Many of them also possess local anesthetic effects. Alkaloids produced by plants have been described as fungicides of phytopathogenic fungi [64,65]. An example of this is the isolation of a piperidine alkaloid from *Piper longum* that has fungicidal activity against some phytopathogenic fungi [66]. Additionally, two alkaloids were isolated from *Chimonanthus praecox* (Japanese allspice) seeds and found to have significant fungicidal activity against five phytopathogenic fungi [67].

In another study, allosecurinine present in root of *Phyllanthus amarus* was found to have *in vitro* fungicidal activity (inhibits fungal spore germination) against five phytopathogenic fungi [64]. Furthermore, the dihydropyrrole isolated from *Datura metel* exhibits in vitro fungicidal effect on *Aspergillus fumigatus*, *A. flavus*, and *A. niger* [68].

Singh et al. (2010) [69] reported that a mixture of quaternary alkaloids isolated from *Argemone mexicana* has antifungal activity (inhibited spore germination) against ten phytopathogenic fungi. This significant in vitro antifungal efficacy of plant alkaloids may lead to their usage by farmers in the field to control some plant fungal diseases.

## 3. Agronanotechnology for Control of Fungal Diseases in Plants

Nanotechnology is a technology of nanoscale (based on materials that have 0.1 to 100 nanometres size) materials with many potential applications in daily life. Nanotechnology highlights the uses of submicron particles, molecules, or individual atoms in biological, chemical, and physical systems [70]. Nanotechnology research involves rediscovery of the biological effects of existing antimicrobial agents by controlling their size to modify their effect. Various inorganic and organic antimicrobial particles of nanosize were used to control bacterial, fungal, and viral pathogens [10,71,72,73].

In recent years, products of nano-fertilizers or nano-pesticides containing nanomaterials have been developed into agricultural practices. Recently, biological materials such as microorganisms, plant extracts, marine organisms, and micro-fluids have been used to synthesize nanoparticles (especially metallic ones) [5,74,75,76]. Nanoparticles bioreducted using primary and secondary metabolites of plant extracts “green synthesis” are the most stable, economic, and eco-friendly nanoparticles [77]. These primary and secondary metabolites of plant extracts can not only promote plant growth, suppress fungal pathogens, and efficiently reduce diseases of crops but can also synthesize eco-friendly nanoparticles via acting as an electron shuttle, besides assisting in the stabilization and reduction of metal ions [78,79]. The following are some examples of nanoparticles that represent green nanotechnology application in fungal management along with their targeted pathogens.

### 3.1. Silver Nanoparticles (Ag NPs)

Silver applications in the field of agriculture have gained momentum in very recent years. Praiseworthy efforts were directed toward the discovery of Ag NPs antimicrobial action against pathogens that infect humans; nevertheless, research has been done to reveal their ability to control phytopathogens. Relatively few reports have explored Ag NPs fungicidal activities against numerous phytopathogenic fungi [80]. Additionally, the mechanism by which silver nanoparticles act as fa ungicidal agent is still unclear. Even after confirming Ag NPs in vitro antifungal activity against various fungi, their application in management of phytopathogenic fungi in the fields continues to be unnoticeable. Research also confirmed that Ag NPs addition to soil or use as coatings for plant seed or seedlings has the ability to control phytopathogenic fungi as well as plant growth promotion.

The green synthesis of silver nanoparticles is the most consistent process for their synthesis [81]. Various researchers reported different sources of plant extracts for the green synthesis of Ag NPs [82]. The first approach involved Alfalfa sprouts for silver nanoparticles synthesis [83]. Ahmad et al. (2012) [84] used the extract of *Punica granatum* peels as the reducing agent to synthesize gold and silver nanoparticles. In another study, *Ziziphora tenuior* leaves extract was used for green synthesis of Ag NPs [85]. Ag NPs synthesized via the green chemistry were also extensively applied as disinfectant, such as in water sanitization [86].

Krishnaraj et al. (2012) [87] used leaf extract of *Acalypha indica* for rapid synthesis of silver nanoparticles and reported their antifungal activity at a concentration of 15 mg against several phytopathogenic fungi such as *Rhizoctonia solani*, *Sclerotinia sclerotiorum*, *Alternaria alternata*, *Botrytis cinerea*, *Macrophomina phaseolina*, and *Curvularia lunata*. Another experiment was conducted to synthesize Ag NPs by mixing AgNO_3_ at concentration of 1 mM with seeds extract of *Thevetia peruviana* at a concentration of 10% and reported their antifungal effectiveness against *Curvularia lunata* (Wakker) Boedijn, which causes leaf spot disease in maize [88]. 

Relatively few studies were conducted on silver nanoparticles used to control fungal diseases in plants in vivo. These studies demonstrated that silver nanoparticles significantly affect the colonial formation of spores of plant pathogenic fungi. Thus, the precautionary application of silver nanoparticles in agriculture may result in the superior efficiency of these nanoparticles due to their direct contact with the spores along with germ tubes of plant pathogenic fungi that suppress fungal viability [87].

### 3.2. Zinc Oxide Nanoparticles (ZnO NPs)

A study was conducted to synthesize inexpensive and eco-friendly zinc oxide nanoparticles by extract of *Parthenium hysterophorus* L. leaves, and demonstrated that these nanoparticles could effectively reduce *Aspergillus flavus* and *Aspergillus niger* growth [89]. Senthilkumar and Sivakumar (2014) [90] used aqueous leaves extract of green tea (*Camellia sinensis*) to synthesize zinc oxide nanoparticles and confirmed their antifungal activity against *Aspergillus fumigatus*, *Aspergillus flavus* and *Aspergillus niger*. Lakshmeesha et al. (2019) [91] reported biofabrication of ZnO NPs using buds extract of *Syzygium aromaticum* flowers and confirmed ability of these nanoparticles to control *Fusarium graminearum* via inhibiting its mycelial growth and mycotoxins production. In another study, zinc oxide nanoparticles biofabricated by *Eucalyptus globules* were proved to exhibit fungicidal effects on pathogenic fungi infecting apple orchards such as *Alternaria mali*, *Diplodia seriata*, and *Botryosphaeria dothidea* [92]. Consequently, these nanoparticles can control fungal diseases and protect fruit crops. 

### 3.3. Gold Nanoparticles (Au NPs)

Green synthesis of antimicrobial Au NPs by diverse extracts of either fresh leaves or flowers of *Magnolia kobus* and *Diopyros kaki* [93], *Azadirachta indica* [94], *Mentha piperita* [95], alfalfa [96], *Helianthus annuus* (sunflower) [97], *Moringa oleifera* [98], and *Artemisia dracunculus* [99] have been described. Additionally, the most frequently used reducing agents for Au NPs synthesis are sodium borohydride and sodium citrate [100].

These Au NPs were also reported to exhibit efficient in vitro antifungal action that can be applied in the field of agriculture to control several phytopathogenic fungi. An example of this is Au NPs synthesized by aqueous extract of *Abelmoschus esculentus* seeds have demonstrated fungicidal effects on *Aspergillus niger*, *Aspergillus flavus*, and *Puccinia graminis* var. *tritci* [101]. In another study, Au NPs synthesized by extract of *Agaricus bisporus* (edible mushroom) and HAuCL_4_·3H_2_O solution have showed higher antifungal effect on *Aspergillus flavus* compared to their effect on *Aspergillus terreus* [102]. Au NPs synthesized by exposure of aqueous gold ions to the leaf extract of *Salix alba* demonstrated more efficient antifungal properties against *Aspergillus niger* and *Alternaria solani*, whereas they showed lower antifungal activity against *Aspergillus flavus*. Moreover, these nanoparticles were relatively unstable at high temperatures [103].

### 3.4. Copper Nanoparticles (Cu NPs)

Green synthesis of copper nanoparticles by leaf extract of Magnolia [104], *Euphorbia nivulia* stem latex [105], *Carica papaya* leaf extract [106], and Aloe Vera leaf extract [107] has been described. Shende et al. (2015) [108] demonstrated green synthesis of these nanoparticles using *Citrus medica* and confirmed their inhibitory effects on *Fusarium oxysporum*, *Fusarium culmorum*, and *Fusarium graminearum*. Therefore, after establishing the in vitro antifungal potentiality of copper nanoparticles against various phytopathogenic fungi, they can be applied in the management of plant fungal diseases [109,110].

## 4. Fungal Cell Deactivation and Evacuation Using Ghost Techniques

Our research group has recently developed a phytopathogenic fungi control trial involving a model of *Aspergillus flavus* and *Aspergillus niger* infection in the tissue culture of the jojoba plant [12]. In this study, we applied the sponge-like protocol for evacuating microbial cells [111,112,113,114,115,116,117] in protecting in vitro tissue cultures of plants against fungal pathogens to establish the use of this protocol and ghost techniques in controlling plant fungal diseases. This work is a step towards a new approach for controlling plant fungal diseases that can be useful for research purposes or may be developed to be introduced in field applications.

The sponge-like protocol was principally created to employ cheap and safe chemical compounds (NaOH, SDS, NaHCO_3_, and H_2_O_2_) for microbial ghost cell preparation for various applications [112]. Ghosts from various microbes were produced by this gentle chemical protocol that was applied to induce evacuation-pores at the microbial cell wall [12,111,112,113,114,115,116,117]. This protocol is unique as it combines the use of minimum inhibition concentration (MIC, which kills microbial cells under minimal killing conditions) and minimum growth concentration (MGC, which allows cells to escape and live but still affects their cell wall) of active chemicals responsible for killing microbes according to the optimal experimental design to prepare microbial ghosts mapped by the full or reduced Plackett–Burman experimental design. 

We successfully prevented plant fungal infection via fungal cell evacuation using the sponge-like protocol. We used the fungal ghost cells formation calculated critical concentration (FGCCC) of active chemicals NaOH, SDS, NaHCO_3_, and H_2_O_2_ to inhibit any form of fungal growth. No sign of fungal growth was observed on the culture medium of treated plants or the plant parts even with multiple spraying steps with fungal cells and FGCCC solutions. On the contrary, negative control experiments which involve plants sprayed with fungal cells only showed visible fungal infection. These findings can be applied through spraying the plants in the field or in tissue cultures with chemical compounds involved in the study at concentrations achieving the best conditions for killing fungal cells and enhancing ghost production. Yet, the optimal incubation time for applying these chemicals to the plant should be extensively screened to prevent any negative effect they may cause on plant growth and quality.

## 5. Other Developing Approaches

### 5.1. Use of Ultraviolet Light to Suppress Plant Fungal Diseases

Plants are continuously suffering from attacks of insects and microbial pathogens. Production of crops with high quality and with minimal input of pesticides at the same time is rather challenging. Thus, alternative approaches like physical treatments that can efficiently suppress diseases in crops is needed. Ultraviolet (UV)-blocking materials like polyethylene nets and films were recently developed to control insect vectors of plant disease which attack greenhouse crops. These materials can filter UV radiation (280–400 nm) and cause interference with insect vision [118]. Roberts and Paul (2006) illustrated that light not only modulates defense responses of plants by influencing plant development and biochemistry, but is also vital for the development of resistance [119]. The physical technique of using UV light was tested for its potential to reduce plant fungal diseases such as powdery mildew on leaves of strawberry and apple [120]. It was found that infection was significantly suppressed by exposing plant leaves to a dose of about 30 mJ/cm^2^ of UV, while no negative effects were observed on plant performance. The regular application of light over time is preferred. This technique involves fungal killing by low doses of UV, whereas plants can tolerate much higher doses.

### 5.2. Effect of Arbuscular Mycorrhizal Fungi on Defense Responses of Plants to Fungal Diseases

Arbuscular mycorrhizal fungi are fungal species that form symbiotic associations with the plant root system. They are found with most horticultural, agricultural, and hardwood crops. Arbuscular mycorrhizal symbioses significantly affect the interactions of the plant with other organisms [121]. Mycorrhizal plants have improved resistance to soil-borne pathogens. On the contrary, the effect of arbuscular mycorrhizal fungi on shoot diseases depends on strategies of the attacker. During formation of mycorrhiza, plant defense responses are modulated, possibly by exchange between pathways of salicylic acid and jasmonate dependent signaling. This can affect plant responses to potential enemies through preparing the plant tissues for a more proficient mechanism of defense activation [122]. The arbuscular mycorrhizal fungus *Glomus mosseae* was reported to induce local or systemic resistance to *Phytophthora parasitica* in tomato roots [123]. It effectively reduces disease symptoms through a combination of local and systemic mechanisms. The local mechanism involves induction of the plant defense-related enzymes chitosanase, chitinase, and beta-1,3-glucanase. On the other hand, a systemic effect of mycorrhizal symbiosis was found. This effect was based on lytic activity of root protein extracts against the cell wall of *Phytophthora*.

### 5.3. Homeopathy and Tea Made from Herbal Flowers for Control of Phytopathogenic Fungi

According to homeopathy, illnesses in living organisms are caused by suppressive processes which act against the vital principle. When suppressive forces are adopted, symptoms are triggered by eliminating everything that affects the vital equilibrium [124,125]. This vital energy disequilibrium leads to diseases in plants, resulting in reduction in yield or death of plants. Nevertheless, application of homeopathic medicine minimizes harmful effects on vital energy and restores equilibrium via stimulating plant defense response. Consequently, plants can resist pathogens and diseases with their own means [124,125,126]. The homeopathic medicines phosphorus and *Calcarea carbonica* were used to control white mold in bean plants [127].

Chamomile tea was used to control damping-off disease in seedlings caused by numerous fungal pathogens. Chamomile flowers are rich source of sulfur, thus their tea kills fungi. Elemental sulfur is an organic fungicide which was used for centuries to kill spores of fungi and to prevent fungal diseases. When a plant is infected, a fungicide cannot cure it. As an alternative, either a fungicide or an organic chemical will prevent disease by killing spores of fungi on plants and on the ground [128].

### 5.4. Mycoviruses to Control the Virulence of Phytopathogenic Fungi

Mycoviruses are viruses infecting fungi, therefore mycoviruses associated with hypovirulence may control fungal diseases. Yet, it is uncertain how mycovirus strains survive in the field, and no mycovirus is applied for crop protection in the field. Zhang et al. (2020) [129] used a previously identified mycovirus to convert Sclerotinia sclerotiorum from a typical pathogen to a beneficial endophytic fungus. Their results revealed that mycovirus downregulates main pathogenicity factor genes expression in *S. sclerotiorum* during infection. They suggest that mycoviruses might impact the origin of endophytism. They proposed an innovative approach for disease control that employs strains infected with mycovirus to enhance crop health and release mycoviruses into the field. Mycovirus-infected strains were found to regulate gene expression of plants responsible for defense, circadian rhythm, and hormone signaling pathways, and this consequently promotes plant growth and disease resistance.

### 5.5. Improving Resistance of Plants to Fungal Diseases

The goal of this approach is to produce plants with increased resistance to diseases. In other words, controlling diseases by the basal resistance of the host, which diminishes the need to apply pesticide. As an example of this alternative strategy, grapevine powdery and downy mildew were controlled via development of resistant varieties against pathogens [130]. Numerous resistance genes against powdery mildew or downy mildew, which have been derived from species of Vitis closely associated with cultivated grapevine, were previously identified. However, many virulent strains of pathogens could overcome several key resistance genes, and such resistance breakdowns were previously reported in grapevine. Since genetic factors of resistance are a restricted resource, and their introduction in a new variety is a costly and long-term process, assessment and enhancement of the durability of resistance is crucial. The assessment of breaking down resistance risk involves three levels. At the level of the plant, a pyramiding strategy is employed to limit this risk and improve the durability of resistance; this strategy aims at combining various genes of resistance in the same variety, using resistance genes that are able to control a wide range of pathogens besides relating multiple mechanisms of defense. At the level of the pathogen, its evolutionary potential should be known [131] to help in evaluating the breakdown resistance risk. At the level of the host-pathogen interaction, cultural practices, environmental conditions, and pathogen fitness penalties [132] help to predict the plant resistance genes’ durability. The Inra-ResDur breeding program used multiple sources (Vitis rotundifolia and some Asian and American Vitis) of resistance to pyramid various factors of resistance in the same cultivar. The genetic analyses revealed molecular markers which were integrated in the breeding process to help to select candidate varieties that carry the desired combinations of genes [130].

Another success in breeding of plants for disease resistance is the durable and broad spectrum control of *Blumeria graminis f*. sp. *hordei*, causing powdery mildew on barley. The susceptibility gene Mildew locus O (Mlo) recessive alleles is responsible for this resistance [133]. Plant resistance was also used as a strategy for reducing aflatoxin accumulation by Aspergillus flavus in maize. Six African-bred maize germplasm lines were produced and evaluated for reduction of aflatoxin the US. Results revealed significant reduction in aflatoxin by these [134].

Breeding of plants for disease resistance was also applied for ornamental plants. Nevertheless, not all ornamental plants have natural resistance to disease; hence, disease control depends on the selection of disease-resistant varieties. Therefore, the elucidation of interactions of the host-pathogen and pathogenicity is crucial to develop innovative approaches for enhancing plant resistance to diseases [135]. Approaches of traditional breeding or genetic engineering by which resistance mechanisms derived from other plant species or pathogens are introduced can develop varieties resistant to diseases [136]. The traditional breeding approaches that introduce natural resistance comprise programs of non-transgenic breeding, for example selection of DNA-based markers with numerous breeding cycles to combine the trait of disease resistance and favorable ornamental features into a single plant genotype. The transgenic approach involves transgenes with tight regulation to introduce specific or broad-spectrum disease resistance into elite ornamental plant genotypes [137]. Resistance breeding in ornamentals is relatively limited, since disease-resistance is considered only through the late phases of the breeding line selection [138]. Genetic mapping of disease resistance in ornamentals is comparatively rare because of the large and complex genomes [139]. The transfer of many genes into ornamentals like those related to pathogenesis, chitinase, glucanase, osmotin, and defensin achieves fungal disease tolerance [140]. Genome editing techniques were also applied to engineer disease resistance in ornamentals [141,142].

Strategies of RNA interference (RNAi) (host-induced gene silencing (HIGS) [143], and spray-induced gene silencing (SIGS) [144]) were confirmed to protect plants from fungal pathogens, and accordingly could be used as an alternative to conventional fungicides. To apply crop protection strategies based on RNAi against phytopathogenic fungi, dsRNA molecules were sprayed on foliage or generated by genetically engineered plants. The design of dsRNA is of great importance, because recognizing a proper gene for silencing in addition to identifying which region of the gene to target are crucial for maximizing efficiency. Uptake of dsRNA is enhanced by different strategies, including using formulations and/or carriers that prevent the degradation of dsRNA by (a)biotic factors and facilitate its translocation. Lastly, determining whether the target fungal pathogen has a functional RNAi machinery is also crucial for success of the control [145].

## 6. Future Perspectives

Healthy, non-toxic, effective, and eco-friendly approaches (green strategies) for controlling fungal diseases in plants should be learned by farmers and agriculturists to protect human and animal health and the biodiversity of soil. These approaches must be applied under in vivo conditions for plant fungal disease control. Research should also give more attention to developing further innovations.

## 7. Conclusions

Extensive analysis of the scientific literature presents biological control of phytopathogenic fungi, microbial fungicides and botanical fungicides, agronanotechnology, fungal cell deactivation and evacuation using ghost techniques, UV-light, arbuscular mycorrhizal fungi, homeopathy and herbal teas, mycoviruses, as well as approaches of traditional breeding and genetic engineering as recent, safe, and effective alternative control means (green strategies of fungal control) to chemical fungicides against plant fungal diseases. Even though some of these technologies are still developing, they represent promising control means that deserve proper introduction to field applications. They protect human and animal health and the biodiversity of soil, and thus may revolutionize the agriculture field to eradicate the damaging effects of chemicals on the environment, as well as on human and animal health.

## Data Availability

Data is contained within the article.

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
