# Peer review of "Recent Approaches towards Control of Fungal Diseases in Plants: An Updated Review"

_jof, 2021, doi:10.3390/jof7110900_

Round 1

Reviewer 1 Report

Manuscript ID: 10.3390

Title: Recent approaches toward control of fungal diseases in plants: an updated review.

Authors: Nawal Abd El-Backy and Amro Abd Al Fattah Amara.

Overview, main impressions and general recommendation:

Context.

Cultivated plants, being sedentary, are subjected to many biotic stresses during their life cycle. Among the phytopathogenic agents, virulent fungi are of particular concern since the vast majority of phytosanitary treatments are fungicides. These chemical fungicides have significant repercussions on the environment as well as on the health of the farmer and even the consumer. Also, the development of new, greener (eco-friendly) crop protection strategies is currently a major issue.

Objective of the review.

In this review, the development of new approaches for controlling fungal diseases in plants is up-to-date classified and discussed. The authors try to argue that effort should be done to bring these recent technologies at farmers’ level.

Main approaches discussed.

In this review the approaches are listed as follows:

  • Biological control of phytopathogenic fungi
    1. The use of antagonistic microorganisms in suppressive soil
    2. Biofumigation (biological soil disinfection)
    3. Microbial control of phytopathogenic fungi: microbial fungicides, parasitism, antibiotics production, competition, plant resistance inducers.
    4. Botanical fungicides
      • Jojoba oil
      • Cinnamaldehyde
      • Milsana
      • Phenolic compounds
      • Essential oils
      • Alkaloids compounds
  • Agronanotechnology for control of fungal diseases in plants
    1. Silver nanoparticles
    2. Zinc oxide nanoparticules
    3. Gold nanoparticles
    4. Copper nanoparticles
  • Fungal cell deactivation and evacuation using ghost techniques

Opinion.

I found the document interesting (to read) and topical on this issue (Journal of Fungi). The reading is rather easy and fluid and the English seems correct to me. However, it seems to me that a certain number of approaches are missing while they are topical and that, from my point of view, they should have been addressed. To this is added, as far as I am concerned, problems with the structuring of certain paragraphs or the choice of examples, which I will argue below. In this context, I recommend major revisions of this review before publication.

Major Comments:

  • Paragraph 2 (Biological control of phytopathogenic fungi):
  • The paragraph begins with the presentation of the use of antagonist microorganisms isolated from suppressive soils. In this part, it seems to me that the case of Fusarium oxysporum would have deserved to be cited (the literature is also rich on this subject).
  • In addition, it does not necessarily appear logical to then have the persistence of biofumigation and then to return to microbial control for which the mechanism of antagonism (described in suppressive soils) is taken up. In this context, I will have proposed to start the paragraph with biofumigation, then to talk about suppressive soils with antagonist microorganisms, then to continue with microbial control (use of living microorganisms).
  • Likewise, paragraph 2.1.4 (botanical fungicides) does not seem to me to be structured in a logical way. Indeed, the existing listing presents an oil, then a specific compound (from an oil), then a plant extract (therefore a mixture of active ingredients), then a class of compounds (phenolic compounds) then we come back to oils etc ... In this context, I will suggest that we see better the logic of presentation by starting from complex extracts (Milsana, jojoba oil and essential oils) to go to the classes of compounds then the pure compounds.
  • Approaches not discussed therefore missing from my point of view:

A certain number of approaches which are developing (or are continuing their development) and which also contribute to reducing the quantities of synthetic phytosanitary inputs used have been omitted from this review. I will cite in particular:

  • UV-light (eg. Diaz and Feres, 2007; Roberts and Paul 2006; Gadoury, 2019)
  • Arbuscular mycorrhizas as microbial control (eg. Azcon-Aguilar et al., 2002; Pozo and Azcon-Aguilar, 2007; Jung et al., 2012)
  • Homeopathy (Toledo et al., 2011) and herbal teas
  • Mycoviruses to control the virulence of pathogenic fungi (eg. Zhang et al., 2020)
  • Plant breeding (varietal selection with the presence of resistance genes against certain fungi; eg. Mekapogu et al., 2021)

In this context, I will suggest that the authors discuss the latter in order to this review become as complete as possible.

Author Response

Dear Editor in chief

Dear Editor

Dear Reviewers

I would like to express our deep gratitude for all valuable comments and suggestions made by the respective reviewers. They helped us a lot to improve the manuscript and make it complete as possible.

Kindly, find below the response to all comments made by the three reviewers point by point.

Reviewer 1.

Comment 1: Paragraph 2 (Biological control of phytopathogenic fungi): In this part, it seems to me that the case of Fusarium oxysporum would have deserved to be cited in suppressive soil

Response: cited in text and references No. 18, 19

Comment 2: In addition, it does not necessarily appear logical to then have the persistence of biofumigation and then to return to microbial control for which the mechanism of antagonism (described in suppressive soils) is taken up. In this context, I will have proposed to start the paragraph with biofumigation, then to talk about suppressive soils with antagonist microorganisms, then to continue with microbial control (use of living microorganisms).

Response: Re-arranged as recommended in comment. 

Comment 3: Likewise, paragraph 2.1.4 (botanical fungicides) does not seem to me to be structured in a logical way. I will suggest that we see better the logic of presentation by starting from complex extracts (Milsana, jojoba oil and essential oils) to go to the classes of compounds then the pure compounds.

Response: Re-arranged as recommended in comment.

Comment 4: Approaches not discussed therefore missing from my point of view: I will cite in particular: UV-light (eg. Diaz and Feres, 2007; Roberts and Paul 2006; Gadoury, 2019)

Arbuscular mycorrhizas as microbial control (eg. Azcon-Aguilar et al., 2002; Pozo and Azcon-Aguilar, 2007; Jung et al., 2012)

Homeopathy (Toledo et al., 2011) and herbal teas

Mycoviruses to control the virulence of pathogenic fungi (eg. Zhang et al., 2020)

Plant breeding (varietal selection with the presence of resistance genes against certain fungi; eg. Mekapogu et al., 2021)

Response: All of them added under title: Other developing approaches

Use of ultraviolet light to suppress plant fungal diseases, references 118-120

Effect of arbuscular mycorrhizal fungi on defense responses of plants to fungal diseases, references 121-123

Homeopathy and tea made from herbal flowers for control of phytopathogenic fungi references 124-128

Mycoviruses to control the virulence of phytopathogenic fungi reference 129

Improving resistance of ornamental plants to fungal diseases references 130-139

with my pleasure

Amro

Reviewer 2 Report

MY REVEISON HAS BEEN CARRIED OUT IN THE ATTACHED PDF

YOU SHOULD IMPROVE THE REVIEW WITH ADDING SOME RECENT ARTICLE AS SPECIFIED IN MY COMMNETS

AND TRY TO REWRITE THE CONCLUSION

Author Response

Dear Editor in chief

Dear Editor

Dear Reviewers

I would like to express our deep gratitude for all valuable comments and suggestions made by the respective reviewers. They helped us a lot to improve the manuscript and make it complete as possible.

Kindly, find below the response to all comments made by the three reviewers point by point.

Comment 1: add and environment

Response: added

Comment 2: They are the same thing

Response: one of them deleted

Comment 3: This phrase is not clear, rewrite it again

Response: rewritten

Comment 4: Delete and

Response: deleted

Comment 5: Delete and

Response: deleted

Comment 6: There is no need of this subtitle, you can start directly with 2.1 as (2) then 2.1, 2.2., ......etc

Response: Done as recommended

Comment 7: This phrase is not clear, rewrite it again

Response: rewritten. 

Comment 8: try to explain and discuss well this information and the results obtained

Response: results were detailed.

Comment 9: Add also the following reference which reported the use of Bacillus mojavensis to antagonize some pathogenic fungi such as M. laxa, M. fructicola, M. fructigena, F. oxysporum, R. solani , B. cinerea, A. ochraceus, P. digitatum, Sclerotinia sclerotiorum, C. acutatum and C. parasitica.

Camele I., et al 2019. Bacillus mojavensis: Biofilm formation and biochemical investigation of its bioactive metabolites. J. Biol. Res. 92 (8296), 39-45.  DOI: 10.4081/jbr.2019.8296.

Response: reference was added No 34

Comment 10: Write Plant Essential oils

Response: written

Comment 11: before these details, add some general information about essential oils and its importance

Response: information added 

Comment 12: The font is different

Response: corrected

Comment 13: Add here the use of origanum vulgare, mentha peperita, salvia offcnalis, thyme vulgare, where there are so common in controlling pathogenic fungi, especially post harvest:

  1. Della Pepa T., et al. 2019. Molecules 24, 2576, 1-16. DOI: 10.3390/molecules24142576.
  2. Elshafie H.S., 2016.  J. Med. Food. 19 (11): 1096-1103.  DOI: 10.1089/jmf.2016.0066.
  3. Elshafie H.S., 2015. J. Med Food 18 (8), 929–934. DOI:  10.1089/jmf.2014.0167

Response: three references added, No.  52-54

Comment 14: Start the paragraph with the biological effect of alkaloids

Response: done as recommended.

Comment 15: it is important also to cite the following papers regarding the use of metal complexes against some fungal pathogens for plants, so you can add here or in  a separate paragraph:

Response: added as references No. 71-73

Comment 16: Please ensure this information, since this article of 2012

Reponse: Replaced with more recent reference

Comment 17: Rewrite conclusion

Response: rewritten 

with my pleasure

Amro

Reviewer 3 Report

First of all, congratulations on the meticulous and collaborative work from which this MS was born.

Contentful and nicely edited MS. Perhaps it would have been worthwhile to present the possibilities of plant conditioning, which is especially important in organic farming, and to deal with a new generation of agents that provide surface sterility with highly oxidative active ingredients.

I would like to suggest improvements below:

- citation problems: ad 163 (Copping 2004)

- letter and character errors and editing errors: ad 221, 222, 224, 226, 313, 328, 410, 436, 438, 541, 468, 542, 548, 564, 660 (+ descriptors and prefixes are not highlighted)

- the citation does not appear to be correct or relevant (ad 240, 282 543)

- foreign language text is written in italics: 505, 522, 643

- the date of the citation is contradictory (ad 245)

incomplete citation: ad 551

- recommended: fungal control

- highlighted according to the rules of the nomenclature (usually in italics and species): ad 409, 418, 419, 422, 432, 443, 444, 447, 448, 449, 451-452, 571, 574, 584-585, 587, 595, 597, 600, 699 , 611, 614, 620, 623, 626, 634, 637, 640, 649, 635, 658, 660, 666

- journals are referred to by full or abbreviated names, it is proposed to standardize.

455, 458, 459, 461, 463 -464, 485, 493-494, 503, 506-507, 509, 512, 518, 525, 535, 537-538, 542, 554,

Author Response

Dear Editor in chief

Dear Editor

Dear Reviewers

I would like to express our deep gratitude for all valuable comments and suggestions made by the respective reviewers. They helped us a lot to improve the manuscript and make it complete as possible.

Kindly, find below the response to all comments made by the three reviewers point by point.

Comment 1: citation problems: ad 163 (Copping 2004)

Response:  corrected by deletion

Comment 2: letter and character errors and editing errors: ad 221, 222, 224, 226, 313, 328, 410, 436, 438, 541, 468, 542, 548, 564, 660 (+ descriptors and prefixes are not highlighted)

Response:  corrected

Comment 3: the citation does not appear to be correct or relevant (ad 240, 282 543)

Response: corrected and replaced with relevant ones.

Comment 4: foreign language text is written in italics: 505, 522, 643

Response: corrected

Comment 5: the date of the citation is contradictory (ad 245)

Response: corrected

Comment 6: incomplete citation: ad 551

Response: corrected

Comment 7: highlighted according to the rules of the nomenclature (usually in italics and species): ad 409, 418, 419, 422, 432, 443, 444, 447, 448, 449, 451-452, 571, 574, 584-585, 587, 595, 597, 600, 699 , 611, 614, 620, 623, 626, 634, 637, 640, 649, 635, 658, 660, 666

Response: corrected

Comment 8: journals are referred to by full or abbreviated names, it is proposed to standardize. 455, 458, 459, 461, 463 -464, 485, 493-494, 503, 506-507, 509, 512, 518, 525, 535, 537-538, 542, 554,

Response: corrected

with my pleasure

Amro

Round 2

Reviewer 1 Report

Dear authors,

I see that you have taken into consideration the suggestions I made and thank you for that.

Consequently, it appears to me that the manuscript is thus more fluid to read and, above all, more complete.

However, from my point of view there are still 2 points to be raised:

(1) I do not understand why in the paragraph which concerns varietal improvement by plant breeding, the authors have restricted themselves to ornamental plants. Indeed, I will have put a more general paragraph, dealing with cultivated plants since this strategy has already been deployed for example in grapevine to fight against downy and powdery mildews (a lot of existing scientific literature on this subject). Authors can also investigate whether other crop plants (annuals or perennials) are subject to this type of strategy to improve their natural resistance to fungal pathogens. 

(2) I thought that the authors would also have spoken, in the new strategies studied, of dsRNA and gene silencing in RNAi-based crop protection strategies (see Secic and Kogel, 2021 for review). Why not have mentioned it, especially since it is targeted against fungal pathogens?

Author Response

Dear Editor in chief

Dear Editor

Dear Reviewers

I would like to express our deep gratitude for all valuable comments and suggestions made by the respective reviewers. They helped us a lot to improve the manuscript and make it complete as possible.

Kindly, find below the response to comments made by the reviewer in second round.

Response to reviewer comments:

Reviewer 1.

Comment 1: I see that you have taken into consideration the suggestions I made and thank you for that. Consequently, it appears to me that the manuscript is thus more fluid to read and, above all, more complete. However, from my point of view there are still 2 points to be raised:

(1) I do not understand why in the paragraph which concerns varietal improvement by plant breeding, the authors have restricted themselves to ornamental plants. Indeed, I will have put a more general paragraph, dealing with cultivated plants since this strategy has already been deployed for example in grapevine to fight against downy and powdery mildews (a lot of existing scientific literature on this subject). Authors can also investigate whether other crop plants (annuals or perennials) are subject to this type of strategy to improve their natural resistance to fungal pathogens. 

Response: As examples of this alternative strategy, grapevine powdery and downy mildew, powdery mildew on barley, aflatoxin in maize were stated in the text to be controlled via development of resistant varieties against pathogens [references No. 130-134].

Comment 2: I thought that the authors would also have spoken, in the new strategies studied, of dsRNA and gene silencing in RNAi-based crop protection strategies (see Secic and Kogel, 2021 for review). Why not have mentioned it, especially since it is targeted against fungal pathogens?

Response: Detailed and references are No 143-145

With my pleasure

Amro Amara